



**Carbon amendment stimulates benthic nitrogen cycling during the**
**bioremediation of particulate aquaculture waste**
Georgina Robinson[1,2,*,#], Thomas MacTavish[3], Candida Savage[3,4], Gary S. Caldwell[1], Clifford
L.W. Jones[2], Trevor Probyn[5], Bradley D. Eyre[6] and Selina M. Stead[1]
[1]School of Natural and Environmental Sciences, Newcastle University, Newcastle, NE1 7RU,
UK.
[2]Department of Ichthyology and Fisheries Science, Rhodes University, Grahamstown 6140,
South Africa.
[3]Department of Marine Science, University of Otago, Dunedin 9016, New Zealand.
[4]Department of Biological Sciences and Marine Research Institute, University of Cape Town,
Rondebosch 7700, Cape Town, South Africa.
[5]Marine and Coastal Management, Private Bag X2, Rogge Bay 8012, Cape Town, South Africa
[6]Centre for Coastal Biogeochemistry, Southern Cross University, Lismore, NSW 2480,
Australia
*Corresponding author. Tel +230 5982 4971; Email address Georgina.Robinson@sams.ac.uk
(G. Robinson)
#Current address: The Scottish Association for Marine Science, Scottish Marine Institute,
PA37 1QA, Oban, UK.
**Abstract:** The treatment of organic wastes remains one of the key sustainability challenges
facing the growing global aquaculture industry. Bioremediation systems based on coupled
bioturbation—microbial processing offer a promising route for waste management. We
present, for the first time, a combined biogeochemical-molecular analysis of the short-term
performance of one such system that is designed to process nitrogen-rich particulate





aquaculture wastes. Using sea cucumbers (*Holothuria scabra*) as a model bioturbator we
provide evidence that adjusting the waste C:N from 5:1 to 20:1 promoted a shift in nitrogen
cycling pathways towards the dissimilatory nitrate reduction to ammonium (DNRA), resulting
in net $NH_4^+$ efflux into the sediment and retention of nitrogen within the system. The carbon
amended treatment exhibited an overall net $N_2$ uptake whereas the control receiving only
aquaculture waste exhibited net $N_2$ production, indicating that carbon supplementation
enhanced nitrogen fixation. The higher $NH_4^+$ efflux and $N_2$ uptake was further supported by
metagenome predictions that indicate that organic carbon addition stimulated DNRA over
denitrification. These findings indicate that carbon addition can provide a means to
successfully bioremediate nitrogen-rich effluents. Longer-term trials are necessary to
determine whether this nitrogen retention is translated into improved sea cucumber biomass
yields.

**Copyright statement**
The authors grant Copernicus Publications an irrevocable non-exclusive licence to
publish the article electronically and in print format and to identify itself as the original
publisher.
**1. Introduction**
Intensive land-based aquaculture produces nitrogen-rich effluent that may
detrimentally impact water quality and other environmental parameters. In conventional
recirculating aquaculture systems (RAS), biological filtration and water exchange are
commonly practiced for nitrogen removal; however, microbial nitrogen removal is limited by
the supply of carbon as an electron donor (Castine, 2013). Carbon supplementation is employed
in a number of treatment technologies to overcome this deficiency (Avnimelech, 1999;Hamlin
et al., 2008;Schneider et al., 2006). The addition of exogenous carbon is a pre-requisite for the
successful operation of denitrifying filters that permanently remove dissolved inorganic
nitrogenous wastes by conversion to dinitrogen gas (Roy et al., 2010). Alternatively, in zero
exchange biofloc systems, carbon to nitrogen ratios (C:N) are increased through the addition
of labile carbon sources to promote ammonia assimilation from the water column by
heterotrophic bacteria (Avnimelech, 1999;Crab et al., 2012). The fundamental difference
between these approaches is the ultimate fate of nitrogen within the system i.e. removal versus
retention. Technological advances are focused on the development of dissimilatory processes



to permanently remove nitrogen from the system as $N_2$ gas, while ecological-based systems,
such as biofloc, aim to re-cycle and re-use nitrogen within the culture system.

The stoichiometric approach taken in C:N amendment in biofloc systems recognises

that carbon and nitrogen cycles are coupled; therefore, the relative elemental abundances
control the rate of nutrient cycling and energy flow within the treatment system (Dodds et al.,
2004;Ebeling et al., 2006). The potential for C:N manipulation in sediment-based aquaculture
effluent treatment systems containing deposit feeders (sea cucumbers) was previously
demonstrated by (Robinson et al., in review), wherein the addition of soluble starch to
aquaculture waste significantly improved sea cucumber growth rate and biomass density.
Furthermore, redox-stratified sediments that harboured predominately heterotrophic microbial
communities also supported higher sea cucumber yields, indicating that predominately
reducing conditions are more favourable for deposit feeder growth (Robinson et al.,
2015;Robinson et al., 2016). Since reducing conditions favour anaerobic respiratory and
fermentative pathways, organic carbon supplementation may stimulate anaerobic bacterial
metabolism by increasing the availability of electron donors and/or substrates for fermentation,
in addition to increasing heterotrophic $NH_4^+$ assimilation (Fenchel et al., 2012;Oakes et al.,

2011).

The C:N ratio affects the quantity of nitrogen released during mineralisation, with a net

release of nitrogen occurring below a threshold of 20:1 (Cook et al., 2007;Blackburn, 1986).
(Robinson et al., in review) hypothesised that C:N manipulation may alter the nitrogen cycling
pathways within the sediment microbial community by mediating a shift from ammonification
(net release) to assimilation (net uptake) of $NH_4^+$ by heterotrophic bacteria; however, the effect
of carbon supplementation on nitrogen cycling was not clearly elucidated. An improved
understanding of how C:N manipulation influences benthic nitrogen cycling is necessary in
order to improve nitrogen assimilation and incorporation into secondary biomass. In the current
study, we applied a coupled biogeochemical-molecular approach to further investigate the
effect of carbon supplementation on nitrogen cycling. Incubation experiments were conducted
to quantify benthic fluxes, while sediment microbial communities were examined using 16S
rRNA gene sequencing. The study aimed to test the hypothesis that increasing the C:N of
particulate aquaculture waste from 5:1 to 20:1 would promote the assimilation of $NH_4^+$ by
heterotrophic bacteria, drive shifts in microbial community composition and result in nitrogen
retention in the culture system.





## 2. Materials and methods

### 2.1 Study site and experimental animals

The study was conducted in a purpose-built bio-secure heated recirculating aquaculture system (RAS) described in (Robinson et al., 2015). The experiment was conducted over a fifteen day period from January 30th (day -1) to February 14th (day 14) 2014 using juvenile sea cucumbers (*Holothuria scabra*) imported from a commercial hatchery (Research Institute for Aquaculture III, Vietnam) on September 5th 2013, that were quarantined and acclimated to the experimental system as described in (Robinson et al., in review).

### 2.2 Experimental design

Three experimental treatments were randomly allocated to 15 incubation chambers with five replicates per treatment. The 'initial' (In) treatment was included to ensure that there were no significant differences between treatments prior to the start of the experiment and as an intial reference point for evaluating the effect of the treatments. The 'no added carbon' treatment (-C) with a C:N of 5:1 received aquaculture waste only (26.8 mg day$^{-1}$ wet weight). The 'added carbon' treatment (+C) received aquaculture waste (26.8 mg day$^{-1}$ wet weight) and carbon in the form of soluble starch (7.5 mg day$^{-1}$ dry weight) to increase the C:N to 20:1 from day zero (Table 1).

### 2.3 Experimental system and rearing conditions

Sediment incubation chambers were established by transferring unsieved $CaCO_3$ builder's sand sourced from a commercial dune quarry (SSB Mining, Macassar, South Africa) into Plexiglas® tubes (25 cm long, 8.4 cm internal diameter) sealed with a polyvinyl chloride (PVC) end cap to a depth of 7.5 cm. The incubation chambers were connected via 4.0 mm air tubing and 4.0 mm variflow valves to a manifold receiving seawater directly from a RAS biofilter (see Robinson *et al.* in review for further details). The water flow rate was 50 mL min$^{-1}$, equivalent to 16.34 exchanges h$^{-1}$. The chamber outflows were routed into a main drainage channel and allowed to flow to waste to prevent soluble carbon sources from entering the RAS. Unsieved $CaCO_3$ was pre-conditioned for four weeks in flow-through tanks prior to its transfer into the chambers. The sediment was allowed to condition and stabilise into redox-stratified layers for 14 days prior to commencement of the experiment. No aeration was provided; however, water was continuously mixed at 60 rpm using a magnetic stirring rod positioned 15 cm above the sediment surface. Stirring rates were just below that which caused sediment re-suspension (Ferguson et al., 2004;Gongol and Savage, 2016).





131   The experimental area was fully shaded from direct sunlight. Light intensity was

132 measured during daylight incubations using a light meter (LX-107, Lutron Electronic

133 Enterprise Co. Ltd, Taipei, Taiwan) positioned 10 cm above each chamber. Additionally, a

134 temperature/light logger (Hobo, UA-002-64, Onset, USA) was placed in an additional chamber

135 positioned in the centre of the experimental treatments. The mean (hours) natural photoperiod

136 was 13.34:10.26 (L:D).

### 2.4 Aquaculture waste and carbon additions

138   The aquaculture waste, used as feed for the sea cucumbers, comprised uneaten abalone

139 (*Haliotis midae*) feed and faeces. It was collected daily from the backwash of a sand filter in a

140 recirculating abalone grow-out system. Samples were sent for organic carbon and total nitrogen

141 content analysis (Robinson et al., in review) and the mean C:N was 5.21:1. Soluble starch

142 (Merck Millipore, Pretoria, South Africa) was used as an additional carbon source to increase

143 the C:N to 20:1. Additions of waste with (+C) or without (-C) added carbon commenced on

144 day zero. The aquaculture waste was mixed into a wet slurry and added daily to the incubation

145 chambers at 16:00 from day zero to day 14 at a concentration of 400 mmol C $m^{-2}\,d^{-1}$.

### 2.5 Experimental timeline

147   Baseline data were collected at the start of the experiment (i.e. day -1), with fluxes

148 measured in all 15 chambers under light and dark conditions. All replicates from the In

149 treatment were sacrificed on day zero and sub-cored for analysis of sediment characteristics.

### 2.6 Sea cucumber growth

151   Animals (n = 30) previously acclimated in the RAS were suspended in mesh containers

152 for 24 h to evacuate their guts prior to weighing and photo-identification (Robinson et al.,

153 2015). Three juvenile *H. scabra* with a mean (± standard deviation) weight of 1.91 ± 0.36 g

154 were added to each of 10 chambers (equivalent to a high stocking density of 1,034.00 ± 12.73

155 $gm^{-2}$) on day zero. They were removed at the end of the experiment (day 14), gut-evacuated

156 for 24 h and reweighed. Wet weight data were used to calculate growth rate (g $d^{-1}$; Robinson

157 *et al*., 2015).

### 2.7 Benthic flux incubations

159   Benthic flux incubations were conducted on day -1 for all treatments (In, -C and +C)

160 and on alternate days from day one to day 13 for the -C and +C treatments, after sacrifice of

161 the In treatment. Light incubations were conducted during daylight hours, commencing





approximately two hours after sunrise (08:00 local time) and dark incubations were conducted
approximately two hours after sunset (22:00 local time). When data were collected the flow
from each chamber was interrupted, the stirrers were paused and the chambers were uncapped
by removing the rubber bung. A portable optical meter (YSI ProODO, YSI Pty Ltd, USA) was
inserted through the sampling port to measure temperature ($\pm$ 0.01 °C) and dissolved oxygen
(DO) concentrations ($\pm$ 0.01 mg L$^{-1}$). The pH ($\pm$ 0.01 pH units) was measured electro-
chemically (Eutech Instruments pH 6+ portable meter, Singapore).
Water alkalinity and nutrient concentration (ammonia, nitrate/nitrite, nitrite and
phosphate) were recorded at the start and end of each light/dark incubation period. To do this,
samples were withdrawn using a 50 mL acid washed plastic syringe connected to the chamber
outflow through 4.0 mm tubing and filtered (Whatman® glass microfiber filters grade GF/C,
Sigma Aldrich, Johannesburg, South Africa) into 15 mL screw-capped polycarbonate vials. All
nutrient samples were immediately frozen at –20 °C and alkalinity samples were kept cold at
4 °C. The N$_2$ samples were taken on three sampling occasions (days one, seven and 13) during
dark incubations, as during daylight hours bubbles may form that interfere with the estimation
of N$_2$:Ar and thus overestimate N$_2$ production (Eyre et al., 2002). To minimise bubble
introduction, N$_2$ samples were collected by allowing the water to flow by gravity from the
chamber outflow directly into 7 mL gas-tight glass vials with glass stoppers filled to
overflowing. The N$_2$ samples were poisoned with 20 µL of 5 % HgCl$_2$ and stored submerged
at 20 °C. The N$_2$ samples were collected in duplicate or triplicate, thus the final values represent
the mean value calculated for each replicate.
After withdrawal of all water samples, replacement water was gravity fed into the
chamber directly from the manifold and the chambers were re-capped and the stirrers re-started.
All materials used for sample collection were acid washed, rinsed three times with distilled
water and air dried prior to use. Total oxygen exchange was measured in three randomly
selected chambers during incubations (one from each treatment) to ensure that the oxygen
concentration did not decrease by more than 20 %. Incubation times were kept short, ranging
from 68 to 146 minutes with an average duration of 104 minutes, to prevent oxygen depletion
and ensure that flux rates were linear (Burford and Longmore, 2001;Glud, 2008).
***2.8 Nutrient analyses***
Dissolved nitrate and nitrite (NOx; 0.01 µM) were determined colourimetrically by
flow injection analysis (QuikChem® 8500 Automated Ion Analyzer, Hach Company, U.S.A.)
and a commercially available test kit (QuikChem® method 31-107-04-1-E for the





determination of nitrate and nitrite in seawater). All other nutrient samples were analysed
manually. Ammonium (0.01 μM) and dissolved inorganic phosphate (0.01 μM) were
determined using the methods of Grasshoff (1976) and Grasshoff et al. (1999) respectively,
and nitrite ($NO_2^-$; 0.01 μM)) was determined according to Bendscheider and Robinson (1952).
***2.9 Gas analyses***
Alkalinity (0.01 mg $L^{-1}$) and total dissolved $CO_2$ (0.01 μM) concentrations were
determined by potentiometric titration according to Edmond (1970) using an automated titrator
system (876 Dosimat plus, Metrohm, USA). Total alkalinity was calculated according to the
method of Snoeyink and Jenkins (1980). $CO_2$ concentrations were calculated from alkalinity
and pH using the equations given in Almgren et al. (1983). Changes in pH and alkalinity were
used to calculate dissolved inorganic carbon (DIC) fluxes.
Dinitrogen gas ($N_2$) was determined from $N_2$:Ar using membrane inlet mass
spectrometry (MIMS) with $O_2$ removal (± 0.01%). Measurement of direct $N_2$ fluxes using this
technique represents the net benthic flux of $N_2$ resulting from a combination of processes that
produce $N_2$, such as denitrification and anammox, and processes that consume $N_2$ such as
nitrogen fixation (Ferguson and Eyre, 2007;Eyre et al., 2013a).
Nutrient and gas fluxes across the sediment-water interface during light and dark
incubations were calculated using initial and final concentration data according to Equation 1.
Net flux rates, representing the net result of 13.57 h of dark fluxes and 10.43 h of light fluxes
were calculated according to Equation 2 (Veuger et al., 2007). Gross primary production was
calculated according to
Equation 3, where light $O_2$ fluxes represent net primary production and dark fluxes
represent respiration. Remineralisation ratios were calculated according to Equation 4 (Eyre et
al. (2013b).
Equation 1     $Flux = \frac{(C_n - C_0) \times V}{A \times t} \times 10,000$
where:
Flux = flux (μmol $m^{-2}$ $h^{-1}$), $C_0$ = concentration at time zero (μmol $L^{-1}$), $C_n$ =
concentration at time n (μmol $L^{-1}$), t = incubation time (h), A = area of sediment surface in
chamber ($cm^2$), and V = volume of water in chamber (L).



Equation 2    Net flux rates $= \dfrac{\text{(hourly dark rates} \times \text{hours of darkness)}+\text{(hourly light rates} \times \text{hours of daylight)}}{24\ h}$
Equation 3    Gross primary production = light $O_2$ flux (+ve) − dark $O_2$ flux (-ve)
Equation 4    Remineralisation ratio $= \dfrac{\text{Dark } O_2 \text{ flux}}{N_2 + NH_4^+ + NO_x}$

***2.10 Sediment sectioning***
On days zero and 14, three sub-cores (internal diameter 30 mm) were extracted from
the In and experimental (-C and +C) chambers respectively. Each sub-core was sectioned into
the following five depth intervals: 0.0 - 0.5, 0.5 - 1.0, 1.0 - 2.0, 2.0 - 4.0 and 4.0 - 6.0 cm for
analysis of sediment characteristics. One set of sub-cores was dried at 50 °C for 24 h for
analysis of total organic carbon and total nitrogen; the second set was frozen in sealed vials in
black bags for spectrophotometric analysis of total carbohydrates. Two sets of samples were
prepared from the third sub-core: sediment samples were frozen in 2 mL Eppendorf tubes for
subsequent deoxyribonucleic acid (DNA) extraction and sequencing. The remaining sediment
was added to 15 mL vials filled with 0.2 µm filtered, one percent buffered paraformaldeyde
and refrigerated for determination of bacterial abundance by flow cytometry.
The organic content measured as particulate organic carbon (OC) and total nitrogen
(TN) was determined on an elemental analyser after removal of carbonates by fuming
(Robinson et al., 2015). Total sediment carbohydrates were measured on defrosted samples
(Robinson et al., in review).
***2.11 Flow cytometry***
Aliquots of preserved samples were prepared in duplicate by staining with
4′,6-diamidino-2-phenylindole (DAPI) for 15 minutes at 4 °C in darkness (Marie et al., 1999).
Bacterial abundance was analysed with a FACSCalibur flow cytometer (BD Biosciences,
Singapore), fitted with a 488 nm, 15 mW laser, using the FL1 detector ($\lambda$ = 530 nm). TruCount
beads (BD Biosciences, Singapore) were used as an internal standard. All cytometric data were
logged and analysed using Cell Quest (Becton-Dickinson) using *Escherichia coli* cells as a
reference. Cell abundance was converted to cells $g^{-1}$ of dry sediment.
***2.12 Deoxyribonucleic acid extraction and importation***
Genomic DNA was extracted from approximately 250 mg of substrate samples using a
DNA isolation kit (ZR Soil Microbe DNA MiniPrep, Zymo Research, USA) yielding purified
genomic DNA for use in polymerase chain reaction (PCR) amplification. Genomic DNA was



stored in sealed, labelled Eppendorf tubes at -20 °C prior to being couriered from the Republic
of South Africa to the United Kingdom. To comply with the Animal Health Act 1981, the
samples were accompanied by a general import license (IMP/GEN/2008/03) for the
importation of animal and poultry products, including DNA, from all non-EU countries.

### 2.13 Polymerase chain reaction and 16S rRNA sequencing

Library preparation was performed using a modified version of the MiSeq WetLab
protocol (Kozich et al., 2013). One microliter of template DNA was arrayed into 96-well plate
format with 17 µL of Accuprime Pfx Supermix (Thermofisher, UK), leaving two wells on each
plate open for controls. Two microliters of reconstituted indexed primers at 100 µM were added
to the samples to barcode them for identification. To identify any contaminating operational
taxonomic units (OTUs), two control samples were included in the sequencing run. The
negative control consisted of one microliter of PCR grade dH$_2$O and the positive control was
one microliter of mock community (HM-278S, BEI Resources, Manassas, USA) at a 1:3
dilution. The primer pair 515F/806R was used to amplify the V4 region of the 16S rRNA gene.
PCR was performed using the following conditions: initial enzyme activation and DNA
denaturation proceeded at 95 °C for two minutes followed by cycling parameters of 95 °C for
20 s, 55 °C for 15 s, 72 °C for five minutes for 30 cycles. A final extension was done at 72 °C
for ten minutes. Amplification of the PCR products was checked on a subset of 12 samples
using gel electrophoresis on a one percent agarose gel prior to library clean up. Samples from
all plates were pooled and libraries were subjected to quality control including quantification
using a KAPA Biosystems Q-PCR kit, obtaining a bioanalyser trace using the Agilent
Technologies HS DNA kit and normalisation using the Invitrogen SequalPrep Plate
Normalisation Kit (Thermofisher, UK). Amplicons were sequenced on an Illumina MiSeq
platform by NU-OMICS (Northumbria University, UK).

### 2.14 Processing of raw sequence data

The raw fastq files were processed using Mothur (version 1.37.0) based on the Schloss
MiSeq SOP with modifications. Raw forward and reverse sequence reads were merged to
create contigs prior to quality filtering. The sequence reads were trimmed using a sliding
window of five base pairs (bp) with an average window quality threshold (Q) of 22 or greater.
Sequences containing an ambiguous (N) base, >8 homopolymers or that had a sequence length
<275 bp were discarded. Quality-filtered sequences were aligned using a custom alignment
created for the variable four (V4) region of the 16S rRNA gene using the Silva database



(version 123; July 2015 release). The reads were screened to include only overlapping regions
(based on alignment positions), pre-clustered (number of differences = 1) and checked for
chimeras using the UCHIME algorithm (Edgar et al., 2011).
Taxons classified as 'Mitochondria', 'Eukaryota' or 'unknown' were specified during
the remove.lineage command. The count.groups command was used to determine the minimum
number of reads per sample for normalisation. To standardise sequencing effort, all samples
were subsampled to 550 using the sub.sample command, to ensure that all replicate samples
from the experimental treatments (+C and -C) were retained. The subsampled OTU table
(shared file) and assigned consensus taxonomy (cons.taxonomy.file) were used in downstream
analyses, including alpha and beta diversity, taxonomic composition and metagenome
predictions of the microbial communities.

### 2.15 Statistical analyses and bioinformatics

Environmental (light, temperature, salinity) and flux rate data for nutrients ($NH_4^+$,
$NO_2^-$, $NO_x$ and $PO_4^{3-}$) and gases (DO, DIC and $N_2$ – night only) collected on day -1 during light
and dark incubations were averaged to provide a mean value per replicate chamber for each
diurnal period respectively. The data were tested for homogeneity of variance and for the
normal distribution of the residuals using Levene and Shapiro Wilk tests. One-way analysis of
variance (ANOVA) tested for differences in the environmental, nutrient and gas flux data
between the In, +C and -C treatments on day -1.
The light, water quality and flux rate data (days 1-13) for nutrients and gases were
averaged to provide a mean value for each replicate incubation chamber. It was not possible to
conduct daytime incubations on day nine due to lowered $O_2$ concentrations in the chambers,
therefore light incubation data represents a mean of six values (days one, three, five, seven, 11
and 13), while the mean dark incubation data were calculated from the full set of seven
incubations. The mean temperature, salinity and mean light, dark and net fluxes of nutrients
and gas fluxes, mean remineralisation ratios and mean gross primary production measured
during the experimental period (days 1-13) were analysed using a Student t-test at alpha <0.05.
Sediment characteristics, including organic carbon, total nitrogen, C:N and bacterial cell
abundance were compared using mixed-model ANOVA with treatment (+C and -C) and
sediment depth as fixed factors. When a significant effect was observed, post hoc comparisons
of means were conducted with a Tukey's honest significant difference test. Differences in *H.*
*scabra* growth rate and biomass density were analysed by Student t-test at alpha <0.05. Data



are presented as mean ± standard error unless otherwise stated. All statistical analyses were
performed in Statistica v.13.

Alpha (within-sample) diversity metrics for the number of OTUs (observed), richness

(Chao 1), abundance-coverage estimator (ACE) and diversity (Shannon, Simpson and Inverse
Simpson) were calculated and visualised in the phyloseq package in R (McMurdie and Holmes,
2013). The diversity metrics were generated by the summary.single command by subsampling
to the lowest number of reads per sample (n = 550) and compared across treatments and
sediment depths using mixed model ANOVA.

Patterns in bacterial community structure between treatments and sediment depths were

visualised using principal coordinates analysis (PCoA) based on a Bray–Curtis dissimilarity
matrix calculated from the OTU table in R. In addition, a non-parametric multivariate analysis
of variance (PERMANOVA) was performed on the community distance matrix based on Bray–
Curtis dissimilarity index to test the null hypothesis that there was no difference in the structure
of microbial communities between treatments (In vs. -C vs. +C) and sediment depth using the
'adonis' function of the vegan package in R (Oksanen et al., 2016).

Mantel correlation tests were performed on dissimilarity matrices of the community

and environmental data to provide an indication of how well microbial community data
corresponded to the environmental data. The environmental distance matrix was calculated as
Euclidean distances computed from a metadata table containing all of the data describing light,
water quality, sediment characteristics and net flux rates for gases and nutrients. The
significance of correlation coefficients was assessed using a permutation procedure. In
addition, the correlation between environmental data and the sediment microbial communities
was determined using the 'envfit' function of the 'vegan' package in R (Oksanen et al., 2016).
Since none of the environmental characteristics were significantly correlated with the microbial
community data, the environmental data were not plotted as vectors on the PCoA ordination.

The Tax4Fun package in R was used to predict the metabolic capacities of the microbial

communities from the 16S rRNA sequences. The fctProfiling option was set to TRUE (default)
to predict the metabolic capacities of the metagenomes based on pre-computed Kyoto
Encyclopedia for Genes and Genomes (KEGG) Ortholog reference profiles (Aßhauer et al.,
2015). Only KEGG Pathways within 'nitrogen metabolism' were retained for analysis. The
KEGG pathway map 00910 for nitrogen metabolism and associated information was used to
extract the KEGG ortholog reference numbers involved in the six fully characterised reactions
listed under 'nitrogen metabolism' (supplementary Table 1). Anaerobic oxidation of ammonia



(anammox) was not included, as although this process is recognised in the KEGG database it
has yet to be assigned to a module or reference profile.

The relative abundance of functional genes predicted from the 16S rRNA sequences

within each ortholog reference profile were summed to provide a mean value for each pathway
module for each replicate sample from all sediment depths sampled in all treatments (n = 45).
The relative abundance of functional genes in the In and experiment treatments was illustrated
by graphically plotting vertical depth profiles and analysed statistically using a mixed-model
ANOVA.

**3. Results**
*3.1 Sea cucumber growth and survival*

Survival of sea cucumbers was 100 % in the +C treatment; however, one replicate

chamber from the -C treatment was terminated on day nine following a period of water column
hypoxia, caused by one animal preventing water exchange by blocking the outflow valve. This
resulted in the mortality of all sea cucumbers in this chamber, reducing the overall survival to
80 %. There was no significant difference between the mean sea cucumber wet weight on day
zero or day 14 between treatments; however, despite the short duration of the experiment the
sea cucumbers in both treatments lost mass (decreasing from $1.91 \pm 0.02$ g to $1.62 \pm 0.03$ g; an
overall mean growth rate of $-0.02 \pm 0.00$ g day$^{-1}$). The biomass density decreased from 1,034.00
$\pm 12.73$ g m$^{-2}$ to 874.97 $\pm 18.31$ g m$^{-2}$, although the initial stocking density was comparable to
the final densities (1,011.46 $\pm 75.58$ g m$^{-2}$) achieved in previous carbon amended cultures
(Robinson et al., in review).
*3.2 Gas and nutrient fluxes*

Benthic fluxes of dissolved oxygen and dissolved inorganic carbon (DIC) can provide

an indication of overall benthic metabolism in response to organic enrichment (Eyre et al.,
2011). There were no significant differences in the light, dark or net fluxes of DO, DIC or $N_2$
between treatments on day -1 ($N_2$ dark only). Sediment oxygen consumption was significantly
higher in the +C incubations throughout the experiment in both light and dark incubations
(Student's t-test; t = -2.87, p = 0.006) resulting in a higher net flux of $-2,905.84 \pm 99.95$ µmol
$O_2$ m$^{-2}$ h$^{-1}$ compared to $-2,511.31 \pm 116.81$ µmol $O_2$ m$^{-2}$ h$^{-1}$ in the -C treatment (Fig. 1a). Fluxes
of oxygen and DIC clearly indicated that the sediment metabolism was net heterotrophic.
During the day, DIC release from organic matter degradation exceeded DIC consumption from
primary production (Fig. 1b). There was an influx of oxygen into the sediment during light and



dark incubations, indicating that respiration dominated over photosynthesis; supported by the
lower gross primary production in the +C treatment (Fig. 1d). There were no significant
differences in the light, dark or net fluxes of DIC with a mean net efflux of 12,732.34 $\pm$
2,031.69 µmol C m$^{-2}$ h$^{-1}$ across the treatments (Fig. 1b). The assumed low rates of
photosynthesis may have been due to shading and from turnover of the microphytobenthos
standing stock due to grazing by sea cucumbers (Glud et al., 2008;Mactavish et al., 2012). In
addition, DIC fluxes were four-fold higher than oxygen fluxes, indicating that the majority of
the organic carbon was oxidised by anaerobic pathways (Burford and Longmore, 2001;Eyre et
al., 2011).

The mean dark $N_2$ flux on days seven and 13 was not significantly different between

treatments (Student's t-test; t = -1.29, p = 0.23; Fig. 1c). Carbon supplementation resulted in a
net $N_2$ influx (-142.96 $\pm$ 107.90 µmol m$^{-2}$ h$^{-1}$), indicating that atmospheric nitrogen fixation
dominated over denitrification and anammox during dark incubations. In contrast, the -C
treatment had a small but positive net $N_2$ efflux (17.33 $\pm$ 36.20 µmol m$^{-2}$ h$^{-1}$), indicating that
nitrogen removal pathways, such as denitrification or anaerobic ammonium oxidation
(anammox), were slightly greater than nitrogen fixation.

There were no significant differences in the dark or net fluxes of any of the nutrients

between treatments on day -1; however, the $NH_4^+$ fluxes during light incubations were
significantly different (one-way ANOVA; $F_{(2, 9)}$ = 12.73, p = 0.002). The In chambers had a
significantly higher $NH_4^+$ efflux of 115.32 $\pm$ 11.43 µmol m$^{-2}$ h$^{-1}$ compared with an influx of -
9.77 $\pm$ 11.82 µmol m$^{-2}$ h$^{-1}$ in the -C treatment. The +C treatment had intermediary values with
a mean $NH_4^+$ efflux of 56.03 $\pm$ 25.54 µmol m$^{-2}$ h$^{-1}$. $NH_4^+$ had the highest flux rate throughout
the experiment (Fig. 2b) with mean efflux significantly higher in the +C chambers during light
incubations compared with the -C treatment (182.25 $\pm$ 120.77 vs. 83.90 $\pm$ 26.70 µmol m$^{-2}$ h$^{-1}$,
t-test; t = 2.93, p = 0.005; Fig. 2b). Sediment-water exchange of $NO_2^-$, $NO_x$ and $PO_4^{3-}$ were
unaffected by carbon addition. Mean fluxes of $NH_4^+$, $NO_2^-$ and $PO_4^{3-}$ were positive irrespective
of diel cycle, indicating net release from the sediment (Fig. 2a-c); however, $NO_x$ fluxes were
variable with opposing trends in light, dark and net fluxes between treatments (Fig. 2d). As
both dissolved oxygen consumption and $NH_4^+$ production were higher in the +C chambers this
indicates an overall increase in benthic metabolism during daylight.
***3.3 Sediment characteristics and remineralisation ratios***

The sediment organic carbon (OC) content decreased over the course of the experiment

(Fig. 3a). The largest decrease was observed at the 1.0 – 2.0 cm and 2.0 – 4.0 cm depth intervals



spanning the oxic-anoxic interface; one of the most active zones of organic matter
mineralisation by heterotrophic microorganisms (Reimers et al., 2013). Vertical profiles of
total nitrogen (TN) and the C:N on days zero and 14 followed a similar trend with the most
marked changes occurring at the 1.0 – 2.0 cm and 2.0 – 4.0 cm depth intervals respectively.
Carbon addition did not affect the OC or TN but sediment depth significantly influenced the
OC (mixed model ANOVA, $F_{(4, 20)} = 3.54$, $p = 0.024$; Fig. 3a) and TN content (mixed model
ANOVA, $F_{(4, 20)} = 3.37$, $p = 0.029$; Fig. 3b), being significantly lower at the 1.0 - 2.0 cm depth
interval with mean values of $0.24 \pm 0.02$ % (OC) and $0.03 \pm 0.00$ % (TN) respectively. This
confirms that the oxic-anoxic interface supported the highest rates of organic matter
mineralisation. In contrast, the deepest sectioned interval (4.0 – 6.0 cm) had significantly
higher OC ($0.51 \pm 0.08$ %) and TN content ($0.07 \pm 0.01$ %) than the shallower intervals. Carbon
addition did not significantly increase the sediment C:N in the +C treatment ($7.90 \pm 0.27$)
compared to the -C treatment ($7.12 \pm 0.24$; mixed model ANOVA, $F_{(1, 20)} = 4.52$, $p = 0.054$;
Fig. 3c). However, carbon supplementation resulted in mean remineralisation ratios (after
exclusion of outliers) of $15.68 \pm 7.43$ that were approximately threefold higher than chambers
receiving aquaculture waste only ($5.64 \pm 4.50$), although the difference was not significant (t-
test; $t = 1.08$, $p = 0.32$). Remineralisation ratios were higher in the +C treatment than the
sediment C:N; a trend that is consistent with nitrogen assimilation by heterotrophic bacteria,
including nitrogen fixation (Eyre et al., 2013b). Conversely, in the -C treatment receiving raw
aquaculture waste at a C:N of 5:1, the remineralisation ratios were lower than the sediment
C:N, indicating net release of nitrogen.

### 3.3 Microbial community analysis and nitrogen metabolism functional gene prediction

A total of 781,701 16S rRNA reads were generated. Four samples from one replicate
of the In treatment were removed during sub-sampling due to a low abundance of reads, and
therefore excluded from further analysis. A total of 780,612 sequences in the 41 samples
remained subsequent to quality control, primer trimming, size exclusion, and removal of
unassigned taxons, mitochondria and Eukaryota.
Neither carbon addition, sediment depth nor the interaction between the factors
(treatment × sediment depth) significantly affected the number of sequences, OTUs (observed
species), community richness (Chao and ACE), or diversity measured as Simpson and Inverse
Simpson indices (mixed model ANOVA; $p < 0.05$; Fig. 4). Sediment depth significantly
influenced Shannon diversity, with the highest diversity of 2.85 recorded in the sediment



surface layer (0 - 0.5 cm) and the lowest (1.54) in the 4 - 6 cm layer (mixed model ANOVA;
$F_{(4, 26)}$ = 3.14, p = 0.031).
Flow cytometry data compared relatively well with the 16S rRNA amplicon sequencing
data. Bacterial abundance (cells $g^{-1}$; Fig. 3e), the number of sequences and OTUs were higher
in the In chambers than the experimental chambers sampled on day 14, presumably in response
to grazing by the sea cucumbers. The number of OTUs decreased from 286.81 ± 128.13 in the
In chambers to 176 ± 65.15 and 181.20 ± 45.90 in the +C and -C treatments respectively.
Overall, the community diversity was low: Shannon diversity = 2.31 ± 0.13, Inverse Simpson
= 5.79 ± 0.51. There was a marked increase in community richness at the 1 - 2 cm depth
interval, coinciding with the oxic-anoxic interface. In the In chambers the number of OTUs
was 778.00 ± 731.00, compared with 343.33 ± 199.25 and 322.67 ± 307.25 in the +C and -C
treatments respectively. The Chao 1 richness indicator also followed this trend (Fig. 4).
The majority of sequences (99.8 %) were assigned to the Bacteria, with only 0.12 %
assigned to Archaea. Taxa from four archaeal phyla were present, including Euryarchaeota,
Thaumarchaeota and Woesearchaeota or were unassigned. *Natronorubrum* (Euryarchaeota)*,* a
halophilic aerobic chemoorganotroph, was the most abundant genus representing 14 of the 27
archaeal reads (Xu et al., 1999).
The bacterial community contained a total of 18 phyla, four candidate phyla and the
candidate division WPS-2. Proteobacteria and Firmicutes were the two dominant phyla
accounting for 47.64 and 34.71 % of the total sequences respectively, with Cyanobacteria
accounting for 7.42 %. Planctomycetes (2.45 %), Actinobacteria (2.34 %), unclassified
Bacteria (2.12 %) and Bacteroidetes (1.33 %) were minor components. The remainder of the
phyla, candidate phyla and the candidate division WPS-2 each represented less than 1 % of the
community. Candidate phyla included Hydrogenedentes (formerly NKB19), Latesbacteria
(formerly WS3), Parcubacteria (formerly OD1) and Poribacteria.
Taxa within the Oxalobacteraceae and the genus *Herbaspirillum* were significantly
more abundant in the -C treatment (Welch's two-sided t-test; p < 0.05; Fig. 5). In comparison,
the genera *Blastopiellula* and *Litorilinea* were significantly enriched in the +C treatment. There
were no significant differences in the mean proportion of taxa between experimental treatments
at phylum, class or order levels, underscoring the high degree of similarity among the microbial
communities between treatments (Fig. 6). Further, there was no correlation between the
microbial community and environmental data (Mantel test; r = 0.04, p = 0.27). The first axis
in the PCoA ordination explained 53.4 % of the variation and appeared to be associated with
sediment depth, while the second axis (4.7 % of the variation) appeared to be associated with




experimental treatment. Treatment did not significantly influence microbial community
structure (PERMANOVA; p<0.05; Table 2), which may be a function of the relatively short
duration of the experiment. By contrast, there was a significant effect of sediment depth on the
microbial community (PERMANOVA; p=0.011; Table 2).
There were no significant differences in the predicted relative abundance of genes
involved in the six nitrogen transformation pathways (mixed model ANOVA; p > 0.05; Fig.
7). The relative abundance of predicted nitrification genes peaked at the 0.5 – 1.0 cm depth
interval in the -C treatment, coinciding with the oxic zone. In the +C treatment, the relative
abundance of predicted denitrification and DNRA genes were higher in the sediment layers
sectioned at 1.0 – 2.0, 2.0 – 4.0 and 4.0 – 6.0 cm. Overall, DNRA was the dominant pathway
(20.52 ± 0.01 %) predicted to occur in all treatments and sediment depths, with the exception
of the surface layer (0.0 - 0.5 cm) in the +C treatment, where there was a higher predicted
relative abundance of denitrification genes (Fig. 7). Denitrification was the second most
abundant predicted pathway (18.02 ± 0.01 %), followed by complete nitrification (8.80 ± 0.43
%), indicating that the potential for coupled nitrification-denitrification was present in all
treatments. Genes predicted to be involved in nitrogen fixation represented 2.85 ± 0.32 %.

## 4. Discussion

Effluent (especially particulates) discharged from intensive land-based aquaculture can
impact the marine benthos through the organic enrichment of the underlying sediment. In this
study, the comparison of vertical sediment profiles before and after the experiment indicated
that the addition of particulate aquaculture waste to treatments with sea cucumbers stocked at
densities of >1 kg m$^{-2}$ did not increase the organic carbon content, total nitrogen or C:N.
Overall, the values were generally lower after 14 days of daily waste addition than at the start.
This is consistent with previous studies that concluded that sea cucumbers are efficient
bioturbators that stimulate benthic microbial metabolism and organic matter mineralisation
(MacTavish et al., 2012).
It was hypothesised that increasing the C:N would mediate a shift from ammonification
(net release) to $NH_4^+$ assimilation (net uptake), leading to an overall decrease in $NH_4^+$ efflux,
however, net $NH_4^+$ production was higher in +C treatments. $NH_4^+$ can originate from four
nitrogen transformation pathways; ammonification (degradation of organic nitrogenous waste),
nitrogen fixation, assimilatory reduction of nitrate to ammonia (ARNA), and dissimilatory
nitrate reduction to ammonia (DNRA), in addition to sea cucumber excretion. ARNA and
nitrogen fixation are both assimilatory pathways that occur within organisms, and therefore do



not contribute to an increase in $NH_4^+$ concentration at the sediment-water interface (Gardner et
al., 2006). Ammonification and DNRA are therefore the only pathways with the potential to
contribute to increased $NH_4^+$ production in the +C treatment, however the increased $NH_4^+$
concentration in the +C treatment is unlikely to have originated from ammonification since the
waste was added on an isonitrogenous basis.

An increasing number of studies have demonstrated the importance, and indeed

dominance of DNRA in nearshore shallow water coastal environments, particularly in tropical
ecosystems (Decleyre et al., 2015;Fernandes et al., 2012;Gardner et al., 2006;Song et al.,
2014;Erler et al., 2013). For example, Fernandes et al. (2012) showed that DNRA can account
for 99 % of nitrate removal in nitrogen-limited mangrove ecosystems. In marine sediments,
DNRA and denitrification compete for nitrate; however, denitrification results in the permanent
removal of nitrogen from the system whereas DNRA retains bioavailable nitrogen in sediments
by reducing nitrate to $NH_4^+$ (Gardner et al., 2006). Since these nitrogen transformation
processes are reductive pathways, mediated by heterotrophic bacteria in the anaerobic zone of
redox-stratified sediments, carbon addition can stimulate both denitrification and DNRA
(Hardison et al., 2015). In some aquaculture systems the availability of organic carbon is
known to limit $N_2$ production via denitrification (Castine et al., 2012); therefore, carbon
supplementation is employed to successfully operate denitrifying filters (Castine, 2013;Roy et
al., 2010). However, Castine (2013) found no significant differences in $N_2$ production when
aquaculture slurries were amended with particulate organic matter or methanol as carbon
sources. Other studies have found that high organic loading rates and/or the addition of
exogenous carbon sources stimulated DNRA and concluded that high organic carbon loading
is a pre-requisite for DNRA to be favoured over denitrification (Hardison et al., 2015;Capone,
2000). In the present study, the higher $NH_4^+$ efflux in the +C treatment, supported by the
metagenome predictions and the influx of $N_2$ gas, would suggest that organic carbon addition
stimulates DNRA over denitrification.

Increasing the organic carbon availability can potentially stimulate all four nitrogen

reduction pathways (supplementary Fig. 1). These pathways, with the exception of
denitrification, result in ammonia production and therefore contribute to nitrogen retention
within the system (Hardison et al., 2015). The factors regulating the balance between
denitrification and nitrogen fixation are not well understood; however, the quality and quantity
of organic carbon may influence the balance between these processes (Fulweiler et al., 2013).
Historically, denitrification has been considered to be the main pathway of nitrogen loss, based
on deficiencies in mass balance calculations (Seitzinger, 1988). However, in sediment-based





systems enriched with particulate organic waste (such as settlement ponds in aquaculture systems) the processes of permanent nitrogen removal account for a very small fraction of the total nitrogen that is permanently removed from the system. For example, Castine et al. (2012) found that denitrification and anammox only removed 2.5 % of total nitrogen inputs to settlement ponds in intensive shrimp and barramundi farms.

Sediment nitrogen fixation can equal or exceed $N_2$ loss in estuarine systems (Newell et al., 2016a). The genetic potential for nitrogen fixation is widespread within the Bacteria and Archaea (Newell et al., 2016b;Zehr and Paerl, 2008a). Heterotrophic nitrogen fixation has not been widely demonstrated in sediments beyond the observation of $N_2$ uptake (Gardner et al., 2006); however, recent studies provide direct evidence by measuring *in situ* $N_2$ production combined with molecular and genomic tools to quantify the presence of the nitrogenase reductase (*nifH*) gene (Newell et al., 2016b;Baker et al., 2015). Indirect evidence for nitrogen fixation is provided in the present study by the presence of *nifH* (K02588) in all samples and the taxonomic composition of the microbial communities.

Nitrogen fixation can be mediated by photoautotrophic and heterotrophic diazotrophs. Heterotrophic diazotrophs, including Gammaproteobacteria and Group A cyanobacteria, are the dominant nitrogen-fixing organisms in oceanic and estuarine systems (Halm et al., 2012;Bentzon-Tilia et al., 2015). In this study, Cyanobacteria was the third most abundant phylum. In the rhizosphere of seagrass beds most nitrogen fixation is mediated by sulphate-reducing bacteria (Welsh et al., 1996). The Deltaproteobacteria, which contains most of the sulphate-reducing bacteria, represented a very small proportion (<0.5 %) of the community; however, Firmicutes were the second most abundant phylum, demonstrating that taxa capable of nitrogen fixation were present (Zehr and Paerl, 2008b).

The addition of exogenous carbon sources including glucose, sucrose and lactose, has been found to stimulate heterotrophic nitrogen fixation in cyanobacteria and sulphate reducing bacteria (Welsh et al., 1997;Newell et al., 2016a). The +C treatment exhibited an overall net $N_2$ uptake whereas the control receiving waste only exhibited net $N_2$ production, indicating that carbon supplementation enhanced nitrogen fixation. Similarly to DNRA and denitrification, the rates of heterotrophic nitrogen fixation in coastal marine sediments are frequently limited by organic carbon availability.

Benthic incubation chambers integrate the exchange of gases and nutrients across the sediment-water interface; thus, while many reactions may be occurring within the sediments, the net outcome of sediment reactions are translated into benthic fluxes. It was anticipated that combining this traditional approach with next generation sequencing would elucidate the



response of sediment microbial communities to carbon addition by highlighting shifts in taxonomy and functional potential. Benthic flux incubations detected a significant enhancement of $NH_4^+$ production during light incubations in response to carbon supplementation; however, no statistically significant differences in the microbial community or predicted nitrogen transformation pathways were observed. Robinson et al. (2016) showed that increasing the availability of rate-limiting electron acceptors (oxygen) had a marked effect on the sediment microbial taxonomic composition, structure, metabolic capacity and functional potential. In contrast, increasing the availability of potential electron donors through carbon supplementation did not significantly affect the microbial community structure. Significant variations at different sediment depths was likely due to the partitioning of processes within the oxic and anoxic layers. None of the environmental parameters, sediment characteristics, and gas or nutrient fluxes were significantly correlated with microbial community structure and no significant differences were observed in the relative abundance of predicted genes involved in the major nitrogen transformation pathways.

The benthic nitrogen cycle is one of the most complex biogeochemical cycles on earth, characterised by a diverse set of dissimilatory microbial processes (Thamdrup and Dalsgaard, 2008). The lack of significant changes in microbial community structure and functioning may indicate that processes that contribute $NH_4^+$ to the sediment were operating concurrently with transformations that removed $NH_4^+$ from the system, such as assimilation by heterotrophic bacteria, anammox and coupled nitrification-denitrification. Furthermore, organic carbon can fulfil many functions under reducing conditions: as an electron donor in redox reactions; a substrate for fermentation; or as an organic substrate assimilated by heterotrophic bacteria coupled with $NH_4^+$ uptake. The resulting effects may have been less discernible than originally predicted.

**5. Conclusion**

Pathways that support retention of nitrogen in sediments can dominate pathways for permanent removal (Newell et al., 2016a), particularly in tropical ecosystems such as seagrass and mangrove systems (the natural habitat of *H. scabra*). This imbalance between denitrification and nitrogen fixation is partially responsible for nitrogen limitation in these systems (Fulweiler et al., 2013;Newell et al., 2016b). Thus, DNRA and heterotrophic nitrogen fixation are important processes for retaining nitrogen and sustaining ecosystem productivity (Fernandes et al., 2012;Enrich-Prast et al., 2016;Decleyre et al., 2015). In shallow euphotic sediments, these processes are likely important for overcoming nitrogen limitation and





competition with benthic microalgae and cyanobacteria, by recycling and retaining $NH_4^+$ in the
sediment. The increase in $NH_4^+$ efflux combined with net influx of $N_2$ into the sediment in
response to carbon addition indicates that even under nutrient loading rates consistent with
hypereutrophic estuaries (400 mmol C $m^{-2}$ $day^{-1}$ and 240 N $m^{-2}$ $day^{-1}$; Eyre and Ferguson,
2009), pathways that retained nitrogen dominated pathways of permanent removal,
underscoring the immense capacity of sediments for assimilating nitrogen from land-based
intensive aquaculture systems.
The coupled biogeochemical-molecular approach was useful in providing an overview
of the functional potential for different nitrogen cycling pathways; however, given the
complexity of nitrogen cycling in marine sediments, future studies should include more
disparate C/N treatments of longer duration and measure all individual processes including
denitrification, anammox, DNRA and nitrogen fixation. Furthermore, the use of more targeted
molecular approaches, such as metagenomic shotgun sequencing or quantitative polymerase
chain reaction (qPCR) in conjunction with stable isotope labelling studies are recommended to
fully elucidate the pathways of nitrogen cycling in response to C:N manipulation.

**Acknowledgements**: This research was funded by a Biotechnology and Biological Sciences
Research Council (BBSRC) Industrial CASE Studentship to G.R. (Grant Code BB/J01141X/1)
with HIK Abalone Farm Pty Ltd as the CASE partner, with additional contributions from the
ARC Grant DP160100248. The work was conceptualised and funding was secured by G.R.,
C.L.W.J., S.M.S, C.S., B. D. E. Experiments were performed by G.R and T.M with equipment
provided by C.S., T.P and B.D.E and data analysed by G.R. The manuscript was written by
G.R. and G.S.C. and edited by and B.D.E, C.L.W.J., C.S., T.M, T.P. and S.M.S. All authors
have approved the final article.

The authors declare no competing financial interests.






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



**Figure legends**

**Fig. 1.** Mean (± standard error) net fluxes (in µmol $m^{-2}$ $h^{-1}$; n = 5) of: a) dissolved oxygen (DO); b) dissolved inorganic carbon (DIC); c) dinitrogen gas ($N_2$); and, d) gross primary production (GPP) in incubation chambers containing sea cucumbers and aquaculture waste with (+C) or without (-C) carbon addition.

**Fig. 2.** Mean (± standard error) benthic light, dark and net fluxes (in µmol $m^{-2}$ $h^{-1}$; n = 5) of: a) phosphate ($PO_4^{3-}$); b) ammonium ($NH_4^+$); c) nitrite ($NO_2^-$); and d) nitrate and nitrite ($NO_x$) in incubation chambers containing sea cucumbers and aquaculture waste with (+C ) or without (-C) carbon addition.

**Fig. 3.** Vertical depth profiles of sediment characteristics: a) organic carbon; b) total nitrogen; c) carbon to nitrogen ratio (C:N); d) total carbohydrate; and, e) bacterial abundance. Cores were sectioned on day zero prior to the addition of aquaculture waste (initial; In) and after waste addition, both with and without carbon supplementation (carbon and no carbon respectively) on day 14.

**Fig. 4.** Alpha diversity metrics calculated on subsampled data. Observed = the number of operational taxonomic units (OTUs); ACE = abundance-coverage estimator; InvSimpson = Inverse Simpson diversity metric.

**Fig. 5.** The mean proportion (%) and the difference in the mean proportion of taxa at: a) family and b) genus level between +C and -C treatments with 95 % confidence intervals. Significant differences in mean proportions were determined using two-sided Welch's t-tests (alpha = 0.05).

**Fig. 6.** Principal Component Analysis ordination of the microbial community structure between the initial (In), +C and -C treatments at the five sediment depth intervals performed on a Bray-Curtis community dissimilarity matrix.

**Fig. 7.** Vertical depth profiles of the predicted relative abundance of genes involved in the six nitrogen transformation pathways: a) nitrogen fixation; b) dissimilatory nitrate reduction to ammonium (DNRA); c) assimilatory nitrate reduction; d) denitrification; e) complete





nitrification; and, f) nitrification, under the pathway module of nitrogen metabolism in the
Kyoto Encyclopaedia for Genes and Genomes (KEGG) database.








**Table 1.** Description of the experimental treatments. The presence (✓) or absence (x) from day
zero of aquaculture waste, added carbon source or sea cucumbers is indicated.

| Treatment | Treatment code | No of replicates | Aquaculture waste | Sea cucumber | Carbon source | C:N |
|---|---|---|---|---|---|---|
| Initial | In | 5 | x | x | x | n/a |
| No added carbon | -C | 5 | ✓ | ✓ | x | 5:1 |
| Added carbon | +C | 5 | ✓ | ✓ | ✓ | 20:1 |






**Table 2.** Results of a non-parametric multivariate analysis of variance (PERMANOVA) testing the differences in microbial community structure at the five sediment depths prior to the addition of aquaculture waste (In) and after waste addition, both with and without carbon supplementation.

| | df | SS | Mean squares | F model | $R^2$ | p |
|---|---|---|---|---|---|---|
| Treatment (T) | 2 | 0.797 | 0.399 | 1.195 | 0.058 | 0.115 |
| Sediment depth (D) | 4 | 1.705 | 0.426 | 1.278 | 0.123 | 0.011 |
| T × D | 8 | 2.656 | 0.332 | 0.996 | 0.192 | 0.494 |
| Residuals | 26 | 8.672 | 0.334 | | 0.627 | |
| Total | 40 | 13.830 | | | 1.000 | |



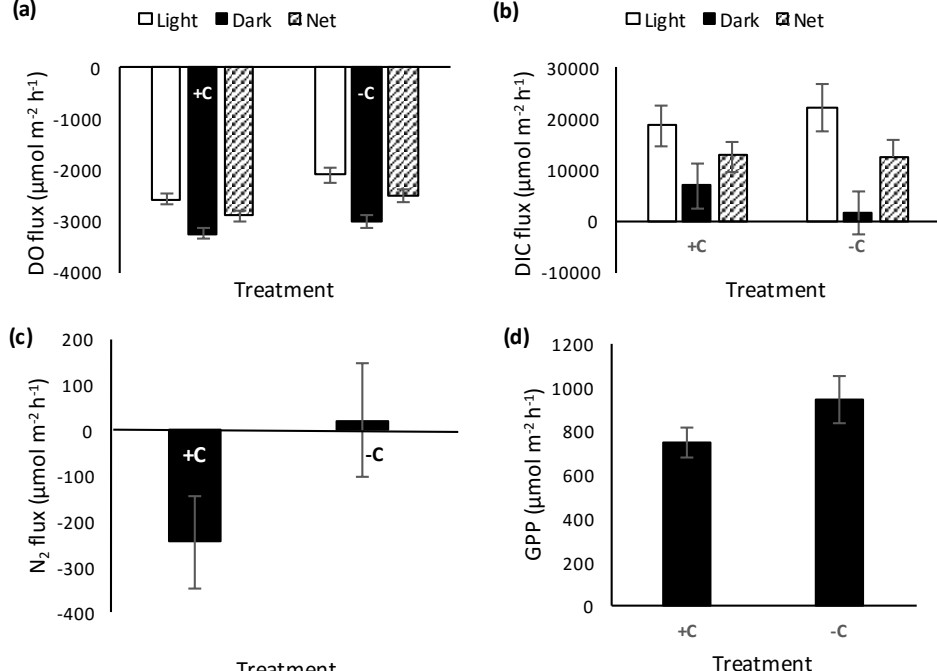

Fig. 1



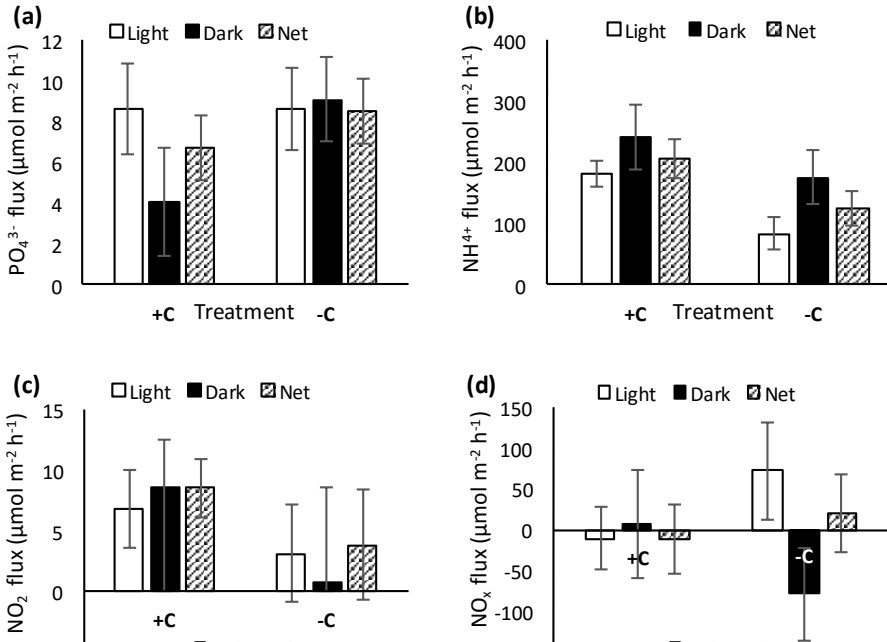

Fig. 2





Fig. 3



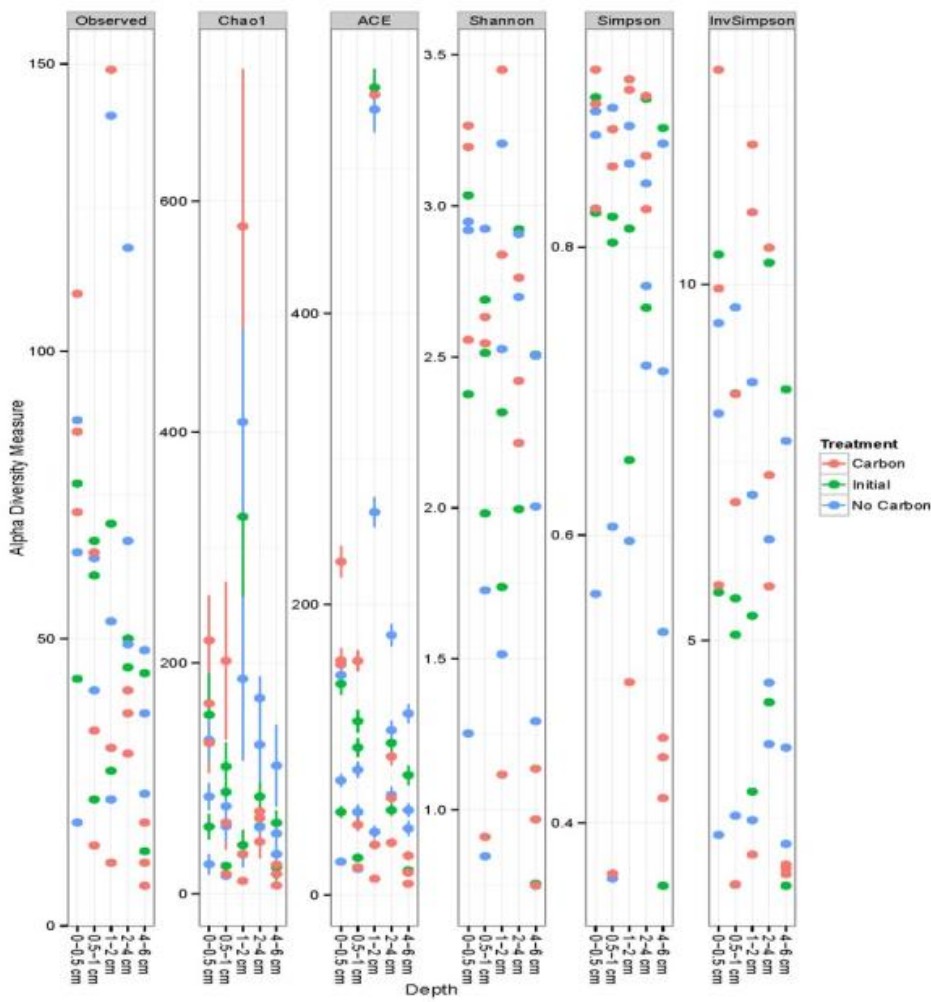

Fig. 4



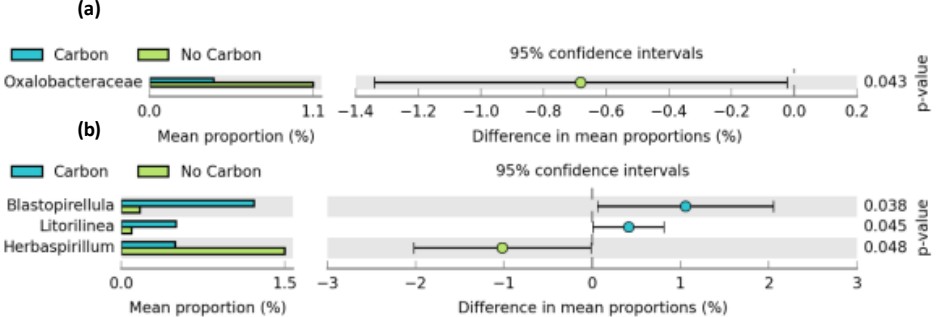

Fig. 5



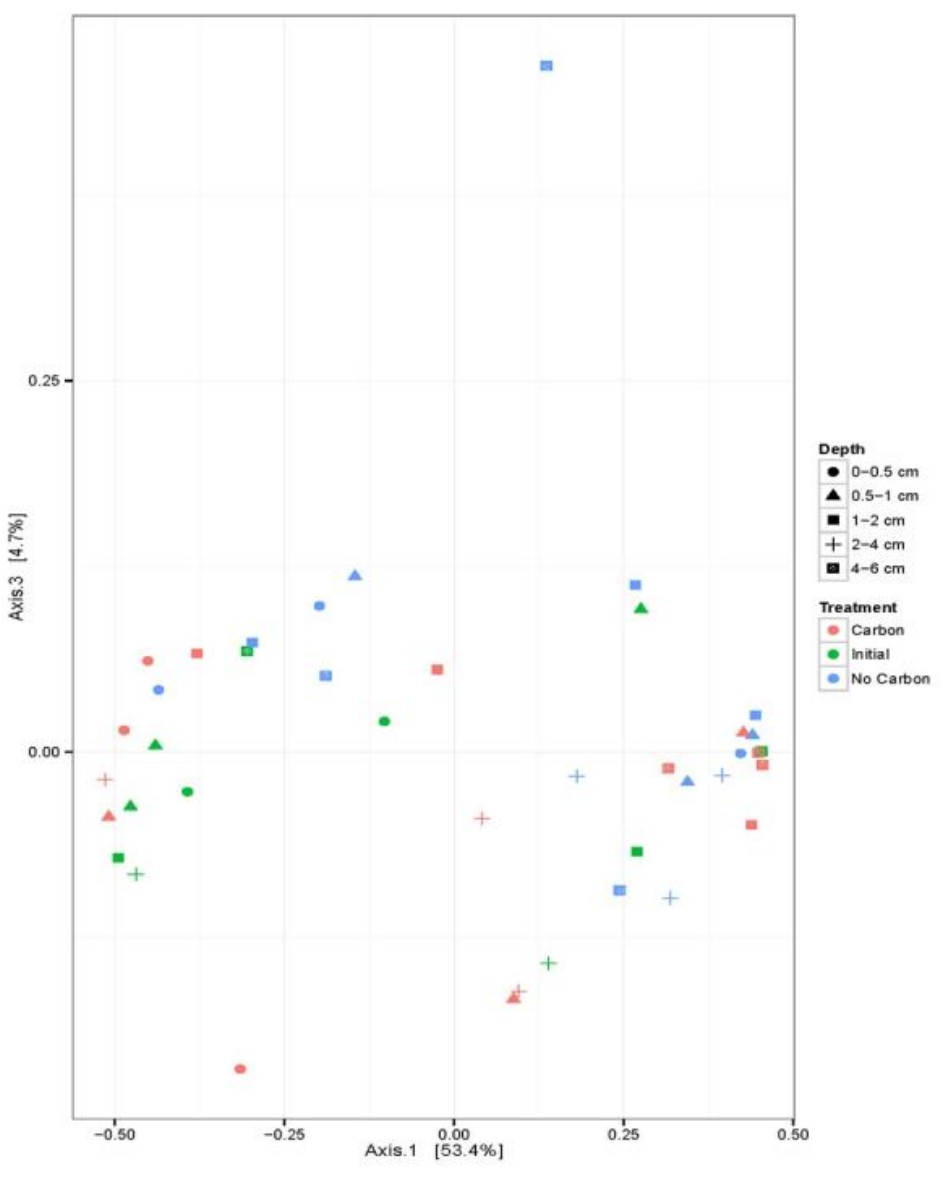

Fig. 6





Fig. 7