# Peer review of "Carbon amendment stimulates benthic nitrogen cycling during the bioremediation of particulate aquaculture waste"

_Biogeosciences, 2017_

## Referee Comment (RC1) · Anonymous Referee #1 · 20 Oct 2017

**Carbon amendment stimulates benthic nitrogen cycling during the bioremediation of particulate aquaculture waste**

**General comments:**
- Good manuscript, highly relevant to understand the fate of metabolic wastes from aquaculture in shallow aquatic systems.
- The manuscript is well structured:
    - The introduction explains well why the hypothesis that increasing the molar C:N ratio of particulate waste from 5 to 20 might promote ammonium assimilation into heterotrophic bacteria is investigated.
    - The material and methods section gives sufficient detail on the set-up of the experimental chambers, the samples collected and the analyses performed, except where C:N ratios are explained (see first major comment).
    - The results section is clearly written and tables and figures are informative (except for giving no information on significant differences between factors/treatments). The latter is taken care of by presenting in-text statistic metrics.
    - The discussion and conclusions are focused around own results and placed in context of existing literature.

**Major comments:**
- Lines 112 – 115: the amounts of aquaculture waste added daily to the incubation chambers are given as '26.8 mg day$^{-1}$ wet weight'. It is not clear if this amount refers to aquaculture waste or to carbon. If it refers to carbon, then it cannot be 'wet weight'. Please clarify.
    - In line 145, it is stated that 400 mmol C/m$^{-2}$/day is added to the incubation chambers. Considering the chambers have an inner diameter of 8.4 cm (Line 119), then about 26.6 mg C/incubation chamber/day is added. This carbon represents dry weight. Please make statements in lines 112-115 and line 145 to concur.
    - Line 114: 'Of soluble starch 7.5 mg DM is added daily'. Here too, it is not clear if this refers to starch or to carbon in starch. Please clarify.
    - Even if above refers to carbon in starch, then the amount is too small to raise the C:N ratio from 5 to 20, assuming the fish waste contains 400 mmol C/m$^2$/day and 80 mmol N/m$^2$/d (= C:N ratio 5). Adding 7.5 mg C per chamber, concurs with 113 mmol C/m$^2$/d. The C:N ratio of the combined fish waste & starch then becomes 6.4. Please clarify.
- Lines 370 – 373: The information that the sea cucumbers lost weight is useful, but comparing to the final weight obtained in similar conditions in another experiment, without giving details on nutrient loading, is not useful. If additional information is given it should give insight why or how the animals lost weight.

**Minor comments:**
- line 84: Start sentence with: The molar C:N ratio...
- whole manuscript: when listing cited references in the text, in some cases, the author names should be written outside the brackets.
- Line 154: a standard deviation is given extrapolating the stocking density from 3 animals per chamber to 541 animals per m$^2$. This cannot be correct.

- Lines 162 and 163: delete 'approximately two hours'. The duration ranges of incubations are given in paragraph lines 183-190.
- Lines 215-216: remove hard return at end of line 215.
- Line 464: delete 'or'.
- Line 466: not clear why a reference is given on an observation of own data?

---

## Referee Comment (RC2) · Anonymous Referee #2 · 24 Oct 2017

General Comments:

Robinson et al. aimed to determine if carbon amendments of aquaculture wastewater would promote nitrogen retention in a sea cucumber culture system. The authors hypothesized that the increased carbon would enhance NH4+ assimilation by heterotrophic bacteria and change microbial community composition to favor nitrogen retention. They found that the amendment did not alter microbial community structure, benthic nutrient fluxes, and sediment characteristics. The carbon amendments did, however, enhance nitrogen fixation suggesting greater nitrogen retention. The manuscript is well-written and aims to address an important research need in the

bioremediation of aquaculture effluent. I also think it provides valuable information on the environmental controls of nitrogen fixation and DNRA. The study results and approaches would be of interest to readers of Biogeosciences, however, I do think the authors need to clarify some of the context of the aquaculture experiment in order for the readers to fully evaluate the study results.

Major Comments:

The authors suggest that improving bioremediation of aquaculture effluent is a study goal. My assumption is that this means increasing nitrogen removal so there is less nitrogen loadings into natural ecosystems. Therefore I find the result of enhanced nitrogen fixation to be conflicting with Lines 43-44 (. . . carbon addition can provide a means to successfully bioremediate nitrogen-rich effluents). I could see fixation and recycling of nitrogen via DNRA being a positive result if the nitrogen was being assimilated by the sea cucumbers. This could then be a removal pathway but that was not measured in this study. Could the authors clarify here? Another way to look at the data set is in terms of a nitrogen budget. Would the carbon amendment result in more nitrogen in the effluent or less?

I appreciate the experimental design and the amount of measurements that were performed in the study. I was surprised by the result of no impact of the carbon addition on sediment carbon content, however, I could see how the sea cucumbers could enhance mineralization. Did the authors consider having treatment(s) with no sea cucumbers? This would have been helpful in interpreting the role of the animals on mineralization/benthic fluxes. For example, how much of the NH4+ efflux is from sediment processes or excretion? Presenting the flux data from the "Initial" trial may help with some of this. Perhaps adding it as a Supplement and including more of this data in the discussion and interpretation of the results. Did the authors run statistical tests comparing Initial, -C, and +C?

I also think it would be helpful to know more about the ambient environmental conditions in the chambers (e.g. nutrients, oxygen, and salinity). The NOx- fluxes into the sediments are low but NH4+ effluxes are high. If NH4+ effluxes are due to DNRA, where is the NOx- coming from? The authors argue that it is not likely due to ammonification (lines 518-421) but they also give data on remineralization ratios that trended higher in the +C treatment (Lines 434-438)?

Minor Comments:

Line 34: "process nitrogen-rich particles" Does "process" imply removal or retention?

Line 40: Consider changing "indicating" to "suggesting"?

Line 74-75: Was the starch treatment a single input or done continuously?

Lines 101-102: Was the system designed to retain nitrogen or remove nitrogen (conventional or biofloc)?

Line 114: A single dose of starch or was it per day?

Lines 151-157: Why did the authors use wet weight instead of dry weight? Why not measure C:N in the sea cucumbers as well?

Line 164: How long were the stirrers paused?

Lines 215-216: Move "Equation 3. . ." to line 215?

Lines 241-242: Can you give a brief description of the carbohydrates method?

Lines 311-313: Did you do any comparisons (ANOVA) with initial, +C, and –C?

Line 397-400: Given the variability (SE) in the N2 fluxes would you want to say that fixation and removal pathways were approximately equal?

Lines 409-410: It would be helpful to know the ambient nutrient concentrations.

Lines 416: Suggests the data is a time-series. Perhaps rewrite as difference between treatments?

Lines 418, 426, 459: This seems like speculation since the oxic-anoxic interface was not measured. Can it be implied with microbial data?

Line 540-542: Consider major comments above.

Line 580: Seems like a reference would be helpful here or are you specifically referring to Welsh 1997 and Newell et al. 2016. Clarify.

Line 610: Consider changing "for" to "over"

Line 623: See major comments above. Is assimilating nitrogen better than nitrogen removal?

---

## Author Response (AR1)

**Response to reviews for Manuscript Number:** bg-2017-293

**Title:** Carbon amendment stimulates benthic nitrogen cycling during the bioremediation of particulate aquaculture waste

**Journal:** Biogeosciences

**MS Type**: Research article

**Authors:** Georgina Robinson, Thomas MacTavish, Candida Savage, Gary S. Caldwell, Clifford L.W. Jones, Trevor Probyn, Bradley D. Eyre and Selina M. Stead.

Dear Editorial Board,

Please find below our point-by-point response to the accurate and helpful comments of the two anonymous reviewers. We thank the reviewers for their prompt and detailed review of our manuscript, which we feel has considerably improved the clarity and accuracy of the methods reported in the manuscript.

A 'track changes' and a clean copy of the revised manuscript are provided to help the reviewers verify that the requested changes have been made, errors have been corrected and suggestions incorporated where possible. Please note that line numbers used here in the response to the reviewers' comments (*in italics*) refer to the line numbers in the track changes version of the revised manuscript.

**REVIEWER 1**

**Major comments**

- **Reviewer 1, Point 1**. Lines 112 – 115: the amounts of aquaculture waste added daily to the incubation chambers are given as '26.8 mg day$^{-1}$ wet weight'. It is not clear if this amount refers to aquaculture waste or to carbon. If it refers to carbon, then it cannot be 'wet weight'. Please clarify.
    o In line 145, it is stated that 400 mmol C/m-2/day is added to the incubation chambers. Considering the chambers have an inner diameter of 8.4 cm (Line 119), then about 26.6 mg C/incubation chamber/day is added. This carbon represents dry weight. Please make statements in lines 112-115 and line 145 to concur.
    o Line 114: 'Of soluble starch 7.5 mg DM is added daily'. Here too, it is not clear if this refers to starch or to carbon in starch. Please clarify.
    o Even if above refers to carbon in starch, then the amount is too small to raise the C:N ratio from 5 to 20, assuming the fish waste contains 400 mmol C/m2/day and 80 mmol N/m2/d (= C:N ratio 5). Adding 7.5 mg C per chamber, concurs with 113 mmol C/m2/d. The C:N ratio of the combined fish waste & starch then becomes 6.4. Please clarify.

*Reviewer 1 has picked up a number of unfortunate errors in the calculations of the elemental ratios (section 2.2) of the aquaculture waste and carbon additions (section 2.4) that have occurred during the editing of the thesis and manuscript. These errors have now been fully*

*corrected and the rationale taken in the study has been more fully explained to facilitate interpretation by readers.*

*The approach taken in the study was to target the upper loading for benthic organisms of 400 mmol C $m^{-2}$ $day^{-1}$ for the treatments that received additional carbon to increase the C:N from 5.21 to 20:1. Thus, 400 mmol C $m^{-2}$ $day^{-1}$ does not refer to the C:N of the fish waste alone, rather it refers to the target C:N of 20:1 to be achieved in the aquaculture waste + carbon (+C) treatments. The overall C:N ratio of 20:1 at 400 mmol C $m^{-2}$ $day^{-1}$ represents the carbon present in the aquaculture waste (104.06 mmol C $m^{-2}$ $day^{-1}$) plus the carbon present in the starch (295.58 mmol C $m^{-2}$ $day^{-1}$). The sentence referring to the equivalent rate of carbon loading has been clarified and moved to the experimental design section.* Lines 119-120 now read: *The carbon addition treatments (+C) were standardised at a concentration of 400 mmol C $m^{-2}$ $d^{-1}$.*

*As the carbon loading was different between treatments, the quantity of aquaculture waste was standardised between treatments at 215.06 mg of wet waste per chamber per day. To calculate the quantity of aquaculture waste and carbon to add, the molarity of carbon was converted into mass of carbon and the C:N of 20:1 is thus expressed on a mass basis and not as a molar ratio. This has been clarified, such that lines 118–119 now read:.....* 'to increase the C:N to 20:1 (mass ratio) from day zero (Table 1)'.

*There was an error in the quantities reported in the previous version of the manuscript. In particular, the 26.8 mg $day^{-1}$ wet weight reported in reference to the aquaculture waste was in fact referring to the target quantity of carbon to be added to achieve a C:N of 20:1 at 400 mmol C $m^{-2}$ $day^{-1}$ for the aquaculture waste plus carbon treatments. During the experiment, the carbon addition treatments received 215.06 mg of wet waste per chamber per day plus 44.50 mg of dry starch per chamber per day. The errors have now been corrected and the manuscript text has been re-written to clarify the volumes of aquaculture waste, starch and their equivalents in mmol of C added. Lines 115-119 now read:* The 'no added carbon' treatment (-C) with a C:N of 5:1 received aquaculture waste only (215.06 mg $day^{-1}$ wet weight). The 'added carbon' treatment (+C) received aquaculture waste (215.06 mg $day^{-1}$ wet weight) and carbon in the form of soluble starch (44.50 mg $day^{-1}$ dry weight) to increase the C:N to 20:1 (mass ratio) from day zero (Table 1).

**Reviewer 1, Point 2**. Lines 370-373: The information that the sea cucumbers lost weight is useful, but comparing to the final weight obtained in similar conditions in another experiment, without giving details on nutrient loading, is not useful. If additional information is given it should give insight why or how the animals lost weight.

Reviewer 1 helpfully pointed out that the rate of nutrient loading in the previous study of Robinson et al. (*in review*) was not reported. This information has been included following the suggestion and Lines 380-383 now read: *The biomass density decreased from 1,034.00 ± 12.73 g $m^{-2}$ to 874.97 ± 18.31 g $m^{-2}$, although the initial stocking density was comparable to the final densities (1,011.46 ± 75.58 g $m^{-2}$) achieved in previous carbon amended cultures standardised at 200 mmol C $m^{-2}$ $day^{-1}$ (Robinson et al., in review).*

*In addition, a new paragraph has been added to the discussion to highlight possible reasons for the difference in growth performance. Lines 627-641 now read:* Our findings indicate that carbon addition may partly bioremediate nitrogen-rich effluent by retaining nitrogen within the system, however longer-term trials are necessary to determine whether this translates into improved sea cucumber biomass yields. In the current study, the sea cucumbers decreased in mass with growth rates of 0.02 g.day-1, however there was no significant difference in mean wet weight of the sea cucumbers at the start or end of the experiment. Two key factors are likely to have accounted for the differences in growth performance of Holothuria scabra in the present study and the previous study of Robinson et al. (in review). Firstly, chambers were shaded from direct sunlight in this experiment to mitigate against water temperature spikes that would likely have caused hypoxia in the small sealed chambers. However, because high light levels may be important for Holothuria scabra growth (Battaglene et al. 1999), this may have resulted in the lower growth performance.Secondly, the duration over which the sediment microbial community was allowed to develop differed between the studies. In Robinson et al. (in review) the trials lasted 112 days compared with the current 28 day study (14 day preconditioning and 14 day experimental).

**Minor comments**

• line 84: Start sentence with: The molar C:N ratio...
*As the C:N ratios used are presented as mass ratios and not molar ratios, this suggestion has not been adopted to maintain consistency in the revised manuscript.*

• whole manuscript: when listing cited references in the text, in some cases, the author names should be written outside the brackets.
*Thank you for highlighting this inconsistency in the citations. All references have now been checked and corrected.*

• Line 154: a standard deviation is given extrapolating the stocking density from 3 animals per chamber to 541 animals per m2. This cannot be correct.
*The standard deviation reported was based on the average weight of all animals for all treatments, however since this is misleading and not accurate, it has been removed.*

• Lines 162 and 163: delete 'approximately two hours'. The duration ranges of incubations are given in paragraph lines 183-190.
*The suggested text has been deleted*

• Lines 215-216: remove hard return at end of line 215.
*The hard return has been deleted*

• Line 464: delete 'or'.
*Here, the sentence has been re-written to improve clarity. There were 3 phlya identified and a number of taxa that were not assigned at phylum level. Lines 478-479 now read:* Taxa from three archaeal phyla were present, including Euryarchaeota, Thaumarchaeota and Woesearchaeota.

• Line 466: not clear why a reference is given on an observation of own data?
*The reference refers to the classification of Natronorubrum, however its placement at the end of the sentence is misleading. Lines 479-481 now read: Natronorubrum* (Euryarchaeota), a halophilic aerobic chemoorganotroph (Xu et al., 1999), was the most abundant genus representing 14 of the 27 archaeal reads.

**REVIEWER 2**

**Major Comments**

**Reviewer 2, Point 1**: The authors suggest that improving bioremediation of aquaculture effluent is a study goal. My assumption is that this means increasing nitrogen removal so there is less nitrogen loadings into natural ecosystems. Therefore I find the result of enhanced nitrogen fixation to be conflicting with Lines 43-44 (: : : carbon addition can provide a means to successfully bioremediate nitrogen-rich effluents). I could see fixation and recycling of nitrogen via DNRA being a positive result if the nitrogen was being assimilated by the sea cucumbers. This could then be a removal pathway but that was not measured in this study. Could the authors clarify here? Another way to look at the data set is in terms of a nitrogen budget. Would the carbon amendment result in more nitrogen in the effluent or less?

*The assumption made by Reviewer 2, that increasing nitrogen removal would fulfill the study goal of improving bioremediation of aquaculture effluent, is perfectly valid, since a general perception in aquaculture bioremediation is that processes that permanently remove nitrogen from the system are beneficial, while processes that result in nitrogen retention are detrimental. It is the opinion of the lead author, however that ecologically-based aquaculture bioremediation systems that aim to re-use and recycle nitrogen, by promoting assimilation by heterotrophic biomass or secondary organisms such as sea cucumbers, may provide a more sustainable approach to the future development of aquaculture bioremediation. This is indeed the subject of an opinion piece "As we see it' recently submitted to Aquaculture Environment Interactions. A new sentence and a reference have been added to reflect and clarify this opinion, lines 69-71 now read:* This study aims to advance ecologically-based aquaculture bioremediation systems that may provide an alternative to closing the nitrogen cycle through the promotion of assimilatory processes (Robinson, *in review*).

The reference 'Robinson, G.: Shifting paradigms and closing the nitrogen loop, Aquaculture Environment Interactions, *in review*' has been added to the reference list.

*As the reviewer points out, however since the amount of nitrogen retained in sea cucumber biomass was not measured in this study, the statement in the abstract has been revised such that lines 43 - 46 now read:* These findings indicate that carbon addition may provide a means to successfully bioremediate nitrogen-rich effluents, however longer-term trials are necessary to determine whether this nitrogen retention is translated into improved sea cucumber biomass yields.

**Reviewer 2, Point 2**: I appreciate the experimental design and the amount of measurements that were performed in the study. I was surprised by the result of no impact of the carbon addition on sediment carbon content, however, I could see how the sea cucumbers could enhance mineralization. Did the authors consider having treatment(s) with no sea cucumbers? This would have been helpful in interpreting the role of the animals on mineralization/ benthic fluxes. For example, how much of the $NH_4^+$ efflux is from sediment processes or excretion? Presenting the flux data from the "Initial" trial may help with some of this. Perhaps adding it as a Supplement and including more of this data in the discussion and interpretation of the results. Did the authors run statistical tests comparing Initial, -C, and +C?

*Reviewer 2 makes a very valid point regarding the consideration of a treatment with no sea cucumbers. The actual experimental design was a fully crossed design with the carbon addition (+C/-C) as one factor and the presence or absence of sea cucumbers (+SC/-SC) included in addition to the initial treatments. However, it was decided to analyse and present this data elsewhere (manuscript in prep.) since the presentation of the full set of results may detract from the study goal of determining the effect of carbon addition on aquaculture waste. Also, the effect of sea cucumbers on the mineralization of aquaculture waste has been previously studied and reported by two of the co-authors* Mactavish, T., Stenton-Dozey, J., Vopel, K. and Savage, C. (2012) 'Deposit-feeding sea cucumbers enhance mineralization and nutrient cycling in organically-enriched coastal sediments', *PLoS One*, 7(11), e50031 [Online].

*Statistical tests (one-way analysis of variance) comparing Initial, -C, and +C on day -1 were run, as explained in lines 303-305 (original pdf of submitted manuscript). The results of these statistical tests were reported in lines 377-378 (original pdf of submitted manuscript) for the gas fluxes and lines 401-406 (original pdf of submitted manuscript) for the nutrient fluxes. However, the helpful suggestion of the reviewer has been adopted and the flux data from the experimental treatments on day -1 has been included in the supplementary material as Fig. S1 (gases) and Fig. S2 (nutrients). The original Fig. S1 has been changed to Fig S3.*

Reviewer 2, Point 3: I also think it would be helpful to know more about the ambient environmental conditions in the chambers (e.g. nutrients, oxygen, and salinity). The NOx-fluxes into the sediments are low but NH4+ effluxes are high. If NH4+ effluxes are due to DNRA, where is the NOx- coming from? The authors argue that it is not likely due to ammonification (lines 518-421) but they also give data on remineralization ratios that trended higher in the +C treatment (Lines 434-438)?

*Following the suggestion of Reviewer 2, the ambient environmental conditions (mean ± standard error) recorded in the incubation chambers on day -1, at the start of light and dark incubations, have been included in the supplementary material as Table S1. The original Table S1 has been modified to Table S2.*

*The comment that Reviewer 2 made regarding the NH4+ effluxes has been fully taken on board and this section of the results have been revised. Lines 535-538 now read:* Ammonification and DNRA are therefore the only pathways with the potential to contribute to increased $NH_4^+$ production in the +C treatment. The increased $NH_4^+$ concentration may have originated from an increase in ammonification conisistent with the increase in metabolism observed in the +C treatment.

**Minor Comments**

Line 34: "process nitrogen-rich particles" Does "process" imply removal or retention?
*The term "process" was used in a neutral sense and could imply permanent removal or retention of nitrogen in the system, however in order to keep the abstract concise and within the word limit, the term 'process' has been changed to 'receive'. Lines 33-36 now read:* We present, for the first time, a combined biogeochemical-molecular analysis of the short-term performance of one such system that is designed to receive nitrogen-rich particulate aquaculture wastes.

Line 40: Consider changing "indicating" to "suggesting"?

*This suggestion has been adopted in Line 41 of the abstract.*

Line 74-75: Was the starch treatment a single input or done continuously?
*The starch was added on a daily basis to the +C treatments, however this had been clarified in the manuscript. Lines 149-152 now read:* Additions of waste with (+C) or without (-C) added carbon commenced on day zero. The aquaculture waste was mixed into a wet slurry while the starch was dissolved in seawater and added daily to the respective treatments at 16:00 from day zero to day 14.

Lines 101-102: Was the system designed to retain nitrogen or remove nitrogen (conventional or biofloc)?
*The experimental system comprised a conventional RAS designed to remove ammonium through conversion to nitrate in the biological system. To clarify, the word conventional has been inserted so that lines 104-105 now read:* The study was conducted in a purpose-built bio-secure heated conventional recirculating aquaculture system (RAS) described in Robinson et al. (2015).

Line 114: A single dose of starch or was it per day?
*The starch addition was done daily, to clarify lines 117-119 now read:* The 'added carbon' treatment (+C) received aquaculture waste (215.06 mg day$^{-1}$ wet weight) and carbon in the form of soluble starch (44.50 mg day$^{-1}$ dry weight) daily to increase the C:N to 20:1 (mass ratio) from day zero (Table 1).

Lines 151-157: Why did the authors use wet weight instead of dry weight? Why not measure C:N in the sea cucumbers as well?
*The wet weight is used in the growth rate calculation, since they were weighed alive at the start and end of the experiment. No sea cucumbers were sacrificed in the experiment, hence the dry weight or C:N ratio of the sea cucumbers was not determined, however this suggestion is useful for future studies.*

Line 164: How long were the stirrers paused?
*The stirrers were interrupted briefly during the start and end of the incubations when data was collected as explained in lines 170-171 and lines 190-193. This has been clarified in the manuscript such that lines 170 to 171 now read:* When data were collected the flow from each chamber was interrupted, the stirrers were paused (~ three min.) and the chambers were uncapped by removing the rubber bung.

Lines 215-216: Move "Equation 3: : :" to line 215?
*This has been done*

Lines 241-242: Can you give a brief description of the carbohydrates method?
The sentence has been re-written to include the name of the method and the reference. *Lines 248-249 now read:* Total sediment carbohydrates (µg g$^{-1}$) were measured using the phenol-sulphuric acid method (Underwood et al., 1995). *The reference has been added to the reference list.*

Lines 311-313: Did you do any comparisons (ANOVA) with initial, +C, and –C?
*This comment has been addressed in the response to Point 2 made by Reviewer 2 in the major comments.*

Line 397-400: Given the variability (SE) in the N2 fluxes would you want to say that fixation and removal pathways were approximately equal?
*Without doing a mass balance, it is not possible to comment on this accurately, however this would be useful in future studies.*

Lines 409-410: It would be helpful to know the ambient nutrient concentrations.
*Following the suggestion of Reviewer 2, the ambient environmental conditions (mean ± standard error) recorded in the incubation chambers at the start of the light and dark periods on day -1, have been included in the supplementary material as Table S1 and referenced in Section 3.2 of the manuscript. Lines 411 – 412 now read:* Ambient environmental conditions recorded in the incubation chambers at the start of the experiment on day -1, during light and dark periods, are presented in Table S1.

Lines 416: Suggests the data is a time-series. Perhaps rewrite as difference between treatments?
*Reviewer 2 makes a very valid point that the phrasing implied time-series data collection. The sentence has been re-written so that lines 429-430 now read:* The sediment organic carbon (OC) content decreased in the experimental treatments after 14 days compared to the initial treatment (Fig. 3a).

Lines 418, 426, 459: This seems like speculation since the oxic-anoxic interface was not measured. Can it be implied with microbial data?
*We have notes recording the position of the level of the oxic-anoxic interface in each chamber as they were sectioned. We have changed the wording to say "approximate depth" (Line 431).*

Line 540-542: Consider major comments above.
*We have amended the manuscript to incorporate all the major comments suggested by this reviewer and thank them for improving the manuscript.*

Line 580: Seems like a reference would be helpful here or are you specifically referring to Welsh 1997 and Newell et al. 2016. Clarify.
*The references of Welsh 1997 and Newell et al. 2016 have been included again here.*

Line 610: Consider changing "for" to "over"
*We have left this unchanged.*

Line 623: See major comments above. Is assimilating nitrogen better than nitrogen removal?
*We feel that we have addressed this comment under the response to point 1 made by Reviewer 2 under Major comments.*

[revised manuscript text omitted]

Fig. 1

[Figure]

Fig. 2

[Figure]

Fig. 3

[Figure]

Fig. 4

[Figure]

Fig. 5

[Figure]

Fig. 6

[Figure]

Fig. 7

**Supplementary material**

**Table S1.** Mean (± standard error) ambient environmental (light, temperature, salinity), nutrient and gas concentrations recorded in the incubation chambers on day -1 at the start of the light and dark incubations.

|  | Light | | | Dark | | |
|---|---|---|---|---|---|---|
|  | **Mean** | | **SE** | **Mean** | | **SE** |
| Light (lux) | 132.08 | ± | 9.63 | - | ± | - |
| Temperature (°C) | 29.34 | ± | 0.06 | 28.62 | ± | 0.04 |
| Salinity (mg L$^{-1}$) | 35.00 | ± | 0.00 | 35.00 | ± | 0.00 |
| pH | 8.03 | ± | 0.00 | 8.24 | ± | 0.00 |
| Ammonia (uM) | 2.93 | ± | 0.13 | 2.58 | ± | 0.23 |
| Nitrite (uM) | 0.29 | ± | 0.08 | 0.58 | ± | 0.09 |
| Nitrate (uM) | 6.98 | ± | 0.56 | 7.46 | ± | 0.51 |
| Phosphate (uM) | 0.57 | ± | 0.03 | 0.47 | ± | 0.01 |
| Dissolved inorganic carbon (uM) | 2,717.56 | ± | 19.90 | 2,357.03 | ± | 27.46 |
| Dissolved oxygen (uM) | 162.60 | ± | 1.06 | 166.28 | ± | 1.04 |
| Nitrogen gas (uM) | - | ± | - | 387.42 | ± | 1.50 |

**Table S2.** Overview of the pathways modules and reference profiles within nitrogen metabolism used to calculate the predicted relative abundance of genes within each pathway. All data was extracted from the Kyoto Encyclopaedia for Genes and Genomes (KEGG) database www.genome.jp/kegg/.

| Pathway | Overview | Module | KEGG Ortholog reference profile (KO) |
|---|---|---|---|
| Nitrogen fixation | Nitrogen => ammonia | M00175 | K02588 + K02586 + K02591 - K00531 |
| Nitrification | Ammonia => nitrite | M00528 | K10944+K10945+K10946 K10535 |
| Denitrification | Nitrate => nitrogen | M00529 | (K00370+K00371+K00374+K00373, K02567+K02568) (K00368,K15864) (K04561+K02305,K15877) K00376 |
| Dissimilatory nitrate reduction | Nitrate => ammonia | M00530 | (K00370+K00371+K00374+K00373, K02567+K02568) (K00362+K00363,K03385+K15876) |
| Assimilatory nitrate reduction | Nitrate => ammonia | M00531 | (K00367,K10534,K00372-K00360) (K00366,K17877) |
| Complete nitrification | Ammonia => nitrite => nitrate | M00804 | K10944+K10945+K10946 K10535 K00370+K00371 |

**Fig. S1.** Mean (± standard error) net fluxes (in μmol m$^{-2}$ h$^{-1}$; n = 5) of: a) dissolved oxygen
(DO); b) dissolved inorganic carbon (DIC); c) dinitrogen gas (N$_2$); and, d) gross primary
production (GPP) in incubation chambers under light and dark conditions on day -1, prior to
the addition of sea cucumbers and aquaculture waste with (+C) or without (-C) carbon.

[Figure]

**Fig. S2.** Mean (± standard error) benthic light, dark and net fluxes (in μmol m$^{-2}$ h$^{-1}$; n = 5) of:

a) phosphate (PO$_4^{3-}$); b) ammonium (NH$_4^+$); c) nitrite (NO$_2^-$); and d) nitrate and nitrite (NO$_x$)

in incubation chambers under light and dark conditions on day -1, prior to the addition of sea cucumbers and aquaculture waste with (+C) or without (-C) carbon.

[Figure]

**Fig. S3.** Nitrogen metabolism pathway map 00910 downloaded from the Kyoto Encyclopaedia for Genes and Genomes (KEGG) database. In the upper part of the diagram, the numbers in the boxes are Enzyme Commission (EC numbers) for enzymes and the chemical reactions they catalyse. In the lower part of the diagram, the enzyme numbers are replaced by the codes for the gene that code for each enzyme. Arrows indicate the direction and pathway of the reactions:

arrows pointing to the right indicate reduction reactions and arrows pointing to the left indicate oxidation reactions. The circles indicate the different inorganic forms of nitrogen.

---

## Author Response (AR2)

Thank you for your revised manuscript. I would like to request further revision in order to avoid an emphasis on conclusions which do not seem fully supported.

Despite your response to reviewer comments, I am not comfortable with the emphasis of this study being on showing that C addition enhances bioremediation. I think that viewing the study through that lens is leading you to make statements that overstretch and are not fully supported by the results. The results presented are that C addition increases both N2 fixation and NH4 flux from the sediment, and it just seems rather contrived to say that these indicate enhanced bioremediation (even if the study were run for longer, and if holothurian biomass were measured, and increased – none of which were actually the case here). I take the argument that the concept of retaining N within the sediment is an alternative form of remediation, but surely NH4+ production does not achieve this (the ammonium would presumably be free to escape and cause eutrophication), and neither does adding N to the system through nitrogen fixation. I therefore think that the conclusion that carbon addition 'may result in greater retention of nitrogen within the system…' is a bit of a stretch, and masks the fact that amendment with C actually leads to N being added to the system.

Nonetheless, the introduction states that C addition is being practised in these new types of system, and that the resultant N cycling is not well understood. I think the manuscript would be much stronger if simply driven by an aim to understand the N cycling changes in response to the C addition practice. Then the full implications of the results can be properly acknowledged, and its impact on bioremediation can be discussed in a fully critical manner, i.e. there could be increased N retention, but there is probably also N addition, and also increased ammonium fluxes, and it is not clear that either of those things is entirely desirable.

Dear Dr Woulds,

Thank you for your insightful comments on our manuscript. We have sought to address these as best we could without compromising the overarching aims of our work. To present a more cautious interpretation of the data we have modified to abstract in two places: Line 43 now reads "These findings indicate that carbon addition may potentially result in greater retention of nitrogen within the system, however longer-term trials are necessary to determine whether this nitrogen retention is translated into improved sea cucumber biomass yields."

Further, we have added an additional two sentences to the end of the abstract that we hope reflect your concerns. "Whether this truly constitutes a remediation process is open for debate as there remains the risk that any increased nitrogen retention may be temporary, with any subsequent release potentially raising the eutrophication risk. Longer and larger-scale
trials are required before this approach may be validated with the complexities of the in-
system nitrogen cycle being fully understood."

We hope that by expressing these points up front in the abstract that we will subtly
shift the weight of interpretation of the paper than will allow the reader to focus more on the
interesting nitrogen pathways that seem to be operating in the system. As you point out, the
nitrogen cycle in these systems is not yet understood (hence the useful contribution that this
study makes).

We have also modified our Conclusion text to provide a more cautious interpretation:

"The increase in $NH_4^+$ efflux combined with net uptake of $N_2$ into the sediment in
response to carbon addition indicates that under nutrient loading rates consistent with
hypereutrophic estuaries (400 mmol C $m^{-2}$ $day^{-1}$ and 240 N $m^{-2}$ $day^{-1}$; Eyre and Ferguson,
2009), pathways that retained nitrogen could dominate over pathways of permanent
removal."

Line 572: Please clarify the following statement 'For example, Castine et al. (2012) found that
denitrification and anammox only removed 2.5 % of total nitrogen inputs to settlement ponds in
intensive shrimp and barramundi farms'. How was the rest of the N removed, if not by a
permanent removal mechanism? What 'counts' as 'removed' (stored in the sediment?).

We have clarified these points by making the following change to the text:

"For example, Castine et al. (2012) found that denitrification and anammox only
removed 2.5 % of total nitrogen inputs (by $N_2$ production) to settlement ponds in intensive
shrimp and barramundi farms. In this case denitrification was not carbon limited; rather the
authors argue that inhibition of microbial metabolism by increased $H_2S$ and $NH^{4+}$ production
limited the performance of the system."

I would be glad to receive another revised manuscript with these comments addressed.

Yours                                                                                                      sincerely,
Clare Woulds

[revised manuscript text omitted]

Fig. 1

[Figure]

[Figure]

[Figure]

[Figure]

Fig. 2

[Figure]

Fig. 3

[Figure]

Fig. 4

[Figure]

Fig. 5

[Figure]

Fig. 6

[Figure]

Fig. 7